# Resveratrol Mitigates Metabolism in Human Microglia Cells

**DOI:** 10.3390/antiox12061248

**Published:** 2023-06-09

**Authors:** Luise Schlotterose, Mariya S. Pravdivtseva, Frowin Ellermann, Olav Jansen, Jan-Bernd Hövener, Frank D. Sönnichsen, François Cossais, Ralph Lucius, Kirsten Hattermann

**Affiliations:** 1Institute of Anatomy, Kiel University, 24118 Kiel, Germany; l.schlotterose@anat.uni-kiel.de (L.S.); f.cossais@anat.uni-kiel.de (F.C.); rlucius@anat.uni-kiel.de (R.L.); 2Section Biomedical Imaging, Molecular Imaging North Competence Center (MOIN CC), Department of Radiology and Neuroradiology, University Medical Center Schleswig-Holstein (UKSH), Kiel University, 24105 Kiel, Germany; mariya.pravdivtseva@rad.uni-kiel.de (M.S.P.); frowin.ellermann@rad.uni-kiel.de (F.E.); jan.hoevener@rad.uni-kiel.de (J.-B.H.); 3Department of Radiology and Neuroradiology, University Medical Center Schleswig-Holstein, 24105 Kiel, Germany; olav.jansen@uksh.de; 4Otto Diels Institute for Organic Chemistry, Kiel University, 24118 Kiel, Germany; fsoennichsen@oc.uni-kiel.de

**Keywords:** resveratrol, neuroinflammation, neurodegenerative diseases, metabolism, M1/M2 microglia

## Abstract

The recognition of the role of microglia cells in neurodegenerative diseases has steadily increased over the past few years. There is growing evidence that the uncontrolled and persisting activation of microglial cells is involved in the progression of diseases such as Alzheimer’s or Parkinson’s disease. The inflammatory activation of microglia cells is often accompanied by a switch in metabolism to higher glucose consumption and aerobic glycolysis. In this study, we investigate the changes induced by the natural antioxidant resveratrol in a human microglia cell line. Resveratrol is renowned for its neuroprotective properties, but little is known about its direct effect on human microglia cells. By analyzing a variety of inflammatory, neuroprotective, and metabolic aspects, resveratrol was observed to reduce inflammasome activity, increase the release of insulin-like growth factor 1, decrease glucose uptake, lower mitochondrial activity, and attenuate cellular metabolism in a ^1^H NMR-based analysis of whole-cell extracts. To this end, studies were mainly performed by analyzing the effect of exogenous stressors such as lipopolysaccharide or interferon gamma on the metabolic profile of microglial cells. Therefore, this study focuses on changes in metabolism without any exogenous stressors, demonstrating how resveratrol might provide protection from persisting neuroinflammation.

## 1. Introduction

Microglia cells are resident immune cells in the central nervous system (CNS) and respond to a variety of trigger signals, e.g., acute damage or infection [1]. Active microglia cells can roughly be divided into two phenotypes: the classical activated/pro-inflammatory M1 or the alternative activated/anti-inflammatory M2 type [2]. M1 microglia cells produce high amounts of reactive oxygen species (ROS) and pro-inflammatory cytokines such as interleukin-1 beta (IL1β). They increase glucose uptake and glycolysis to achieve a faster and more efficient metabolism. This effect is often referred to as a “metabolic switch” in activated microglia cells [3]. In contrast, M2 cells rely on oxidative phosphorylation, rather than aerobic glycolysis [4]. A broad spectrum of subphenotypes M2a-c can be differentiated, displaying differences in their biochemistry and physiology [1,5]. Neutral microglia, often called resting microglia, are described by the M0 phenotype. This phenotype is maintained by inhibitory signals within the CNS. However, microglia cells are rarely found to be resting, since they always patrol the CNS [6].

Neuroinflammation is the innate immune response in the CNS against harmful and toxic stimuli [7]. Activated microglial-mediated neuroinflammation plays a major role in the pathology of different neurodegenerative diseases. Hence, vast amounts of M1 microglia are found in the brains of Alzheimer’s disease patients and in acute conditions such as traumatic brain injury (TBI) or an ischemic stroke [8,9,10]. Targeting pro-inflammatory M1 microglia to induce a shift to the M2 phenotype could be an effective approach to inhibit neuroinflammation. Thus, triggering metabolic reprogramming and mitigating metabolism represents a potential clinical intervention strategy.

The antioxidant resveratrol (3, 4′, 5 trihydroxystilbene) is known to support the microglia M2 phenotype and inhibit neuroinflammation [11,12]. Resveratrol is a polyphenol found in different plants, produced as a reaction to injury or as protection against fungi. Examples of food with particularly high concentrations of resveratrol are red wine or peanuts. In 1982, resveratrol was first recognized for its cardioprotective effects. Since then, it has appeared as a top hit in different pharmacological screenings, among others for cyclooxygenase inhibitors or activators of sirtuin deacetylases [13]. Today, resveratrol is a candidate for the treatment of numerous diseases, and it is known for its anticarcinogenic, cardioprotective, and neuroprotective effects [14]. The interest in resveratrol is reflected by 200 clinical trials listed on clinicaltrials.gov (https://clinicaltrials.gov/ct2/results?cond=&term=Resveratrol&cntry=&state=&city=&dist=, accessed on 11 May 2023).

In this study, we investigated the effect of resveratrol on the metabolism of human microglia cells. In health and disease, microglia cells are of key interest. Targeting their metabolism in order to change their immunological phenotype is a promising route to combat neurological diseases.

## 2. Materials and Methods

### 2.1. Chemicals and Reagents

Dulbecco’s modified Eagle medium (DMEM) (#41965) and penicillin-streptomycin (#15140122) were purchased from Thermo Fisher Scientific, Darmstadt, Germany. l-glutamine (#56-85-9), methanol HPLC-grade (#7342.1), and chloroform (#3313.2) were ordered from Carl Roth, Karlsruhe, Germany. Fetal bovine serum (FBS) (#P30-3306) was obtained from PAN-Biotech GmbH, Aidenbach, Germany. D_2_O (#STBJ4154), and resveratrol (#R5010) was ordered from Sigma-Aldrich/Merck, Taufkirchen, Germany. Resveratrol was dissolved in polyethylene glycol 400 (PEG400) obtained from Caesar & Loretz GmbH, Hilden, Germany.

### 2.2. Cell Culture

The HMC3 human microglia cell line (Cat# CRL-3304, RRID: CVCL_II76) was purchased from American Type Culture Collection (ATCC, Manassas, VA, USA). Cells were grown in DMEM supplemented with 10% FBS, 1% penicillin-streptomycin (10,000 U/mL), and 2 mM additional l-glutamine, and they were incubated at 5% CO_2_/37 °C. Cells were routinely checked for mycoplasma contamination using mycoplasma-specific PCR (#11-1100, Venor^®^ GeM Classic; Minerva Biolabs^®^, Berlin, Germany).

### 2.3. Inflammasome Activity Assays

For the quantification of inflammasome activity, microglia (HMC3) were seeded in white 96-well plates (8000 cells/100 µL medium/well). Upon 6 h of resveratrol (100 µM) treatment, inflammasome activity was measured in technical duplicates using the Caspase-Glo^®^ 1 Inflammasome Assay (#G9951, Promega, Madison, WI, USA) according to the manufacturer’s instructions. Plates were read using the TECAN GENios microplate reader (Tecan Group Ltd., Männedorf, Switzerland).

### 2.4. Scanning Electron Microscopy (SEM)

Microglia (HMC3) were seeded on 24 mm × 12 mm coverslips that were placed in 6-well plates (60,000 cells/1 mL medium/well). After 24 h resveratrol (100 µM) treatment cells were washed with PBS and fixed for 30 min in 3% glutaraldehyde. In the next step, samples were washed with PBS and kept in a 2% osmium solution for 20 min. Subsequently, all water was removed by placing samples in increasing ethanol concentrations (30–100%), and critical point drying was carried out using a CPD 030 (Bal-Tec, Balzers, Liechtenstein). Finally, samples were coated with an SCD 050 sputter coater (Bal-tec, Balzers, Liechtenstein) for 50 s and imaged using a JSM-IT200 (JEOL, Freising, Germany).

### 2.5. Insulin-like Growth Factor 1 Enzyme-Linked Immunosorbent Assay

The concentration of IGF-1 in the cell supernatant was measured using Human IGF-I/IGF-1 DuoSet ELISA (#DY291, R&D systems, Minneapolis, MN, USA). For the measurement, microglia (HMC3) were seeded in 6-well plates (80,000 cells/1 mL medium/well) and treated with resveratrol (100 µM) for 24 h. Afterwards, cell supernatants were collected and processed according to manufacturer instructions.

### 2.6. Proliferation

Proliferation was determined by counting the microglia (HMC3) seeded in 6-well plates (80,000 cells/1 mL medium/well) using the T20 Automated Cell Counter (Bio-Rad, Feldkirchen, Germany) after 24 h of resveratrol (100 µM) treatment. Proliferation was calculated as an n-fold amount of the initially seeded cell number.

### 2.7. Glucose Uptake Quantification

For analysis of glucose uptake, microglia (HMC3) were seeded in white 96-well plates (5000 cells/100 µL medium/well). Upon 24 h of resveratrol (100 µM) treatment, glucose uptake from the medium was measured in technical duplicates using the non-radioactive Glucose Uptake-Glo™ Assay (#J1341, Promega) according to the manufacturer’s instructions. Plates were read using the TECAN GENios microplate reader.

### 2.8. TMRE-Mitochondrial Membrane Potential Assay

Mitochondrial activity was investigated using the TMRE-Mitochondrial Membrane Potential Assay Kit (#ab113852; Abcam, Cambridge, UK) according to manufacturer’s instructions. Microglia (HMC3) were seeded in 24-well plates (7500 cells/250 µL medium/well). Imaging was carried out upon the 24 h resveratrol (100 µM) treatment using the Keyence BZx800 Fluorescence Microscope (KEYENCE GmbH, Neu-Isenburg, Germany). The fluorescence intensity of two areas from each experiment was quantified using *ImageJ* [15].

### 2.9. ^1^H NMR Spectroscopy Sample Preparation

Microglia (HMC3) were seeded in T75 flasks (3,000,000 cells/15 mL medium/flask) and treated with resveratrol (100 µM) or without resveratrol as controls for 24 h. Subsequently, cells were washed three times with PBS and scraped off in ice-cold methanol. Next ice-cold chloroform and deionized H_2_O were added and mixed intensely with the cells for 30 s., followed by gentle movement on a plate shaker for 10 min. The chloroform/ H_2_O extraction was repeated once. Finally, the upper aqueous phase was transferred to a glass vial and lyophilized using a Lyovac GT2 lyophilizer and freeze dryer (Leybold, Dresden, Germany). Samples were kept on ice for the entire procedure and stored at −80 °C for later retrieval. For ^1^H nuclear magnetic resonance (NMR) spectroscopy, the lyophilized samples were reconstituted in a 2 mM pH 7.03 phosphate buffer in D_2_O. Trimethylsilyl-tetradeuterosodium propionate (TMSP) was added to serve as an internal standard for metabolite concentrations and as a chemical shift reference.

### 2.10. ^1^H NMR Spectroscopy Acquisition

All ^1^H NMR spectra were obtained using a 500 MHz NMR spectrometer (Avance NEO, Bruker, Ettlingen) with a ^1^H, ^19^F, X-TBO probe head at 298 K. For each sample, a one-dimensional ^1^H NMR spectrum, with water suppression using excitation sculpting (zgesgppe, Bruker) [16,17], was acquired using the following parameters: 32 dummy scans, 2680 scans, 90° flip angle, and 4 s repetition time including 2 s acquisition time of 28,570 points, resulting in 14.28 ppm spectral width. One spectrum was recorded in 3 h. Free induction decay was zero filled to 131,072 (128 k) points. Exponential apodization with a line broadening parameter of 0.5 Hz was applied. The full width at the half maximum of TMSP was 0.75 Hz before line broadening and after zero filling.

### 2.11. ^1^H NMR Spectroscopy Processing

The metabolite peaks in ^1^H NMR spectra were identified based on chemical shifts and signal multiplicity in comparison with the literature [18,19] and NMR metabolomic databases (*HMDB* and *BMRE*) [20,21,22,23,24,25]. Then, the most intensive ^1^H NMR peaks observed in the experimental spectra were matched to the corresponding individual metabolites derived from databases.

A multipoint baseline correction was applied to all spectra before quantifying the metabolites. The identified multiplets were fitted within *MestReNova* V14.2.0 (Mestrelab Research, Santiago de Compostela, Spain). The resulting fitted intensity of each multiplet or part of a multiplet was used to quantify the relative concentration of metabolites. If more than one of the non-overlapping multiplets corresponded to the same metabolite, then the average of the integrated intensities was used for the evaluation. The signal of each metabolite was normalized by the fit of the full spectra calculated from 11 to −1 ppm. The normalization process was performed to eliminate any differences in the initial number of cells in each sample. In this way, we obtained relative concentrations for each sample. For each sample group, three independent cell cultures were used for sample preparations and ^1^H NMR spectroscopy analysis, displaying good reproducibility (Appendix A).

### 2.12. Statistical Analysis

All results are presented as mean value ± standard deviation (SD). Statistical analysis was performed using *GraphPad Prism* V9.4.1. Statistically significant differences were evaluated by one-way analysis of variance (ANOVA), followed by Tukey’s post hoc test for comparisons between multiple groups. Probabilities were considered statistically significant at values of *p* < 0.05.

## 3. Results

### 3.1. Resveratrol Minimizes Endogenous Inflammasome Activity in Human HMC3 Microglia and Stimulates IGF-1 Production

In the first step, the effect of resveratrol on endogenous inflammasome activity was investigated. Human HMC3 microglia were treated with 100 µM resveratrol for 6 h at a commonly used concentration [26,27,28]. Afterwards, the activity of caspase-1, an essential component of the inflammasome, was measured. Inflammasomes contribute to the response towards infectious agents and physiological aberration. However, they release pro-inflammatory cytokines such as IL1β and IL18, which can prolong and exacerbate neuroinflammation [7,29]. The resveratrol treatment decreased inflammasome activity by almost three-fold in comparison to the untreated control (*p* < 0.01) (Figure 1A). Thus, resveratrol has the potential to lower the endogenous levels of inflammasome activity. Despite the changes in inflammasome activity, the treatment with 100 µM resveratrol for 24 h had no effect on cell proliferation (Figure 1B). Additionally, microglia cell morphology was not altered upon the application of 100 µM resveratrol (Figure 1C).

To further examine the protective effects of resveratrol on human HMC3 microglia, the concentration of insulin-like growth factor 1 (IGF-1) was measured in the cell supernatant. IGF-1 has protective effects on neurons and is involved in the regulation of TNFα during neuroinflammation [30]. The ELISA of IGF-1 showed a trend towards higher IGF-1 concentrations after 24 h of resveratrol (100 µM) treatment compared to untreated controls (Figure 1B). Therefore, resveratrol stimulates microglia cells to produce neuroprotective factors in addition to general anti-inflammatory effects.

### 3.2. Human HMC3 Microglia Reduce Glucose Uptake and Mitochondrial Activity after Resveratrol Treatment

Next, the glucose consumption of human HMC3 microglia was analyzed by comparing untreated controls to samples supplemented with resveratrol (100 µM) for 24 h. As shown in Figure 2, glucose uptake for cells treated with resveratrol was reduced by nearly 50% compared to the amount consumed by controls (*p* < 0.05).

To delineate the effects of resveratrol on mitochondrial function (Figure 3), changes in the membrane potential were investigated using a tetramethylrhodamine-ethyl ester (TMRE)-based method. TMRE accumulates in active mitochondria due to their negative charge. Once mitochondria become less active, they depolarize, and fewer TMREs accumulate in the mitochondria. After 24 h of treatment with resveratrol (100 µM), the observed fluorescent intensity significantly decreased (*p* < 0.01) in comparison to untreated controls, indicating that resveratrol drastically reduced mitochondrial activity. Nevertheless, the measured fluorescence was still three-fold higher than the fluorescent intensity of the ionophore uncoupler of oxidative phosphorylation carbonyl cyanide 4-(trifluoromethoxy) phenylhydrazone (FCCP) (Figure 3B). Therefore, mitochondria were still alive and functioning after the resveratrol treatment.

The reduction in glucose uptake and mitochondrial activity shows how broadly resveratrol can slow down metabolism.

### 3.3. Resveratrol Attenuates Metabolism in Human HMC3 Microglia Cells

To identify how the 24 h resveratrol (100 µM) treatment affects metabolic pathways, we analyzed the metabolic response in whole-cell extracts from human HMC3 microglia using ^1^H NMR spectroscopy, which comprises the resonance lines of hydrogen atoms in all metabolites and all soluble small-molecule content of the cell.

Figure 4 shows a typical ^1^H NMR spectrum of one control sample. Nine metabolites displaying profound changes compared to resveratrol-treated samples were identified. The endogenous metabolites identified from human HMC3 microglia cells and their resonance assignations are listed in Appendix A.

The quantification of metabolites after the resveratrol treatment revealed significant changes in metabolite concentrations compared to untreated controls (Figure 5). The concentrations of essential amino acids valine (Val, *p* < 0.0001), isoleucine (Ile, *p* < 0.0001), phenylalanine (Phe, *p* < 0.001), and tyrosine (Tyr, *p* < 0.0001) were all found to be significantly increased compared to the untreated controls—by 40% or more. Moreover, the concentration of fumarate—which is a product of tyrosine and phenylalanine depletion [31,32]—was reduced by half compared to untreated controls (Fum, *p* < 0.0001). The metabolite concentration of non-essential amino acids glycine (Gly, *p* < 0.0001) and alanine (Ala, *p* < 0.001) also increased by 30%. Glutamate, in contrast, showed a 20% lower concentration after resveratrol treatment (Glu, *p* < 0.01).

Apart from amino acids, inosine monophosphate (IMP), which occupies a central position in purine metabolism [33], was detected in a 20% higher concentration than in the untreated controls (*p* < 0.05).

Taken together, the metabolic profile of HMC3 microglia is distinctively altered upon the resveratrol treatment, particularly affecting amino acid and purine metabolism.

## 4. Discussion

In the presented study, we investigated the effects of resveratrol on human HMC3 microglia cells. To our knowledge, this is the first time that the effects of resveratrol were analyzed in human microglia. Microglia cells are the immune cells of the brain and become activated upon contact with external stimuli. Once activated, microglia cells switch to a faster metabolism to orchestrate a pro-inflammatory response that aims to protect the brain. However, long-term neuroinflammation can have disastrous consequences ranging from impaired cognition to a loss of synapses and resulting neurodegeneration [2]. In this study, resveratrol demonstrates the ability to attenuate metabolism in human HMC3 microglia cells and induces a more bioenergetically efficient and anti-inflammatory microglia phenotype. These findings highlight the regulatory potential of microglia reactivity for therapeutic purposes, especially in the prevention and treatment of neurodegenerative diseases.

Previous studies have shown the protective effects of resveratrol on neuroinflammation in in vitro or in vivo mice models. Some reported functions were restored cell viability, lowered levels of pro-inflammatory cytokines, and increased levels of protective cytokines in murine BV2 microglia cells and also in vivo after lipopolysaccharide (LPS) or LPS/interferon gamma (IFNγ) treatment [34,35,36]. Furthermore, resveratrol reduced LPS/adenosine triphosphate (ATP)-induced NLRP3 inflammasome activity in murine N9 microglia cells [37]. Nevertheless, previous studies focused on exogenous stimuli such as LPS, IFNγ or ATP, leaving the endogenous effects of resveratrol aside. Our own experiments showed that resveratrol reduced inflammasome activity in human HMC3 microglia cells without additional stimuli. Thus, resveratrol has the potential to not only protect human microglia cells from exogenous stressors but also lower the endogenous levels of inflammasome activity.

According to previous publications, the effects of resveratrol are concentration-dependent. The concentration of 100 µM resveratrol, as used in this study, is classed to be rather high. In high concentrations, resveratrol was found to induce cytotoxicity even for very robust cancer cells, as comprehensively reviewed by Mukherjee et al. [38] and Madreiter-Sokolowski et al. [39]. Nevertheless, in the present study, microglia proliferation was not affected by these high concentrations, which might be a very beneficial aspect in approaches that develop and study glial-friendly local delivery systems [12]. Accordingly, differentiated human THP-1 macrophages tolerate 100 µM of resveratrol very well, regardless of whether it is cis- or trans-resveratrol [40]. In contrast, undifferentiated THP-1 monocytes were much more sensitive to resveratrol than after differentiation to macrophages [41]. Subsequently, different cell types react to very different resveratrol concentrations.

In addition to the reduced inflammasome activity, resveratrol increased the secretion of neuroprotective factor IGF-1. IGF-1 directly protects neurons, inhibits blood–brain barrier permeability, and promotes the protective M2 microglia phenotype [30,42]. Similar results were previously reported by Basta et al. [43], indicating that resveratrol treatment leads to higher mRNA levels of IGF-1 in uninephrectomized rats.

Inflammasomes are more active in the activated/pro-inflammatory M1 microglia phenotype [7,44]. Furthermore, activation increases the energy demand of microglia cells, hence inducing a metabolic switch to extensive glucose uptake and enhanced glycolysis in order to produce higher levels of ATP [45,46]. Thus, inflammation is correlated with faster glycolytic metabolism [4].

Kueck et al. [47] already showed that resveratrol inhibited glucose metabolism in human ovarian cancer cells, and Massimi et al. [26] found resveratrol treatments to induce a decreased conversion of glucose and amino acids in HepG2 cells. In accordance with these two studies, resveratrol was found to lower the amount of glucose consumption from the medium in comparison to untreated controls.

Resveratrol is putatively produced by various plants to inhibit bacterial attacks. Therefore, its effects might be pronounced in mitochondria as they are of bacterial origin according to the endosymbiotic theory [39]. Indeed, after treatment with resveratrol, human HMC3 microglia cells indicated decreased mitochondrial activity. Mitochondrial activity and oxidative stress are associated with different neurodegenerative diseases [48]. A review by Jardim et al. [49] revealed that resveratrol protected mitochondrial function and dynamics, e.g., by decreasing the production of ROS. In contrast to the present study, resveratrol was also found to trigger mitochondrial biogenesis. However, this observation may have been caused by longer experimental time frames and continuous resveratrol supplementation.

It was found that resveratrol treatment decreases the conversion of amino acids, which is in agreement with reduced mitochondrial activity. Amino acids are involved in different metabolic pathways in human cells, e.g., energy generation in the TCA circle in mitochondria [48]. Essential amino acids and non-essential amino acids were also detected in elevated concentrations. Additionally, fumarate, being a product of tyrosine and phenylalanine depletion, was measured in lower concentrations than in untreated controls. These results support earlier findings that state that resveratrol leads to increased levels of amino acids in HepG2 cells [26]. Interestingly, a study by J.-J. Yan et al. [19] revealed that the activation of microglia cells with LPS resulted in a decrease in glycine amino acid. Therefore, the increased concentrations of amino acids measured in resveratrol-treated samples indicate an inactivated microglia phenotype.

In the same study [19], the glutamate concentration was found to be increased after LPS treatment with respect to the post-treatment effects of resveratrol, which is an observation that is opposite to the results shown in this work. Glutamate is a major excitatory neurotransmitter in neuronal and glial cells of the CNS. After a brain injury, glutamate increases to toxic levels, causing cell death by calcium-overload-induced calpain activation. Therefore, glutamate supplementation can be used as a model to mimic the conditions after TBI [50].

Apart from amino acids, the concentration of IMP was elevated after resveratrol treatment. IMP is one of the key molecules in purine metabolism and nucleotide synthesis. Increased levels might indicate a slower synthesis of GMP and AMP. Furthermore, a recent study found that IMP suppressed TNFα production and in turn induced IL10 expression in endotoxemic mice [33].

These results indicate a metabolic adaptation by resveratrol in the opposite direction relative to the metabolic switch detected in activated/pro-inflammatory M1 microglia cells. Therefore, lowering endogenous levels of inflammasome activity and metabolism might protect microglia cells from activation and shifting to the M1 phenotype.

In persisting neuroinflammation, the shift from M1 microglia to M2 microglia is missing. Fewer M2 microglia not only lead to a lack of inflammation control but also to lower levels of neuroprotective factors such as IGF-1 or brain-derived neurotrophic factor (BDNF) [2]. Moreover, neurotoxic reactive A1 astrocytes are induced by M1 microglia cells, displaying a direct link between glial cell activations [51]. The results from this study show how resveratrol induced anti-inflammatory properties by boosting IGF-1 secretion and attenuating metabolism.

It is important to recognize the potential limitations of this study. Despite the use of a human cell line rather than rodent cells, immortalized cell lines come with biological and functional limitations. Microglia in culture are known to display a baseline activation, but this applies also to a part of patrolling microglia in vivo. Additionally, cell cultures are simplified models of the brain, and further work needs to be carried out for transitions to in vivo situation This applies especially to the resveratrol concentration used, which needs to be studied at least in an ex vivo brain approach prior to animal studies.

Although, changes for nine metabolites in the cell lysates were identified and most of them showed a substantial (double digit) and significant change in response to treatment, higher fields or cryogenically cooled probes would allow increasing the sensitivity so that more metabolites may be detected. Of course, using cell lysates is an invasive process that provides a snapshot of the metabolic status at the given time only. The real-time metabolism of living cells and their response to resveratrol may be detected, e.g., using thermal or hyperpolarized NMR [52].

Furthermore, there are limitations in using resveratrol as a potential drug. Resveratrol is metabolized rapidly; therefore, the maintenance of sufficient concentrations in brain tissue is difficult for longer time periods [14]. Potentially, this drawback can be overcome by developing more stable resveratrol analogues [53]. An application via local drug-releasing implants [54] in the case of surgery-associated inflammation or by brain-specific drug delivery systems (e.g., intranasal administration [55]) is feasible as well.

## 5. Conclusions

In summary, this study shows how resveratrol supports the switch to an anti-inflammatory phenotype of human HMC3 microglia cells by mitigating metabolism, decreasing endogenous inflammasome activity, and increasing IGF-1 production. Recognizing the importance of M1/M2 dynamics and the development of drugs such as resveratrol, which assists in the shift to a reduced metabolic and less inflammatory state, will facilitate the treatment of neurodegenerative diseases in the future.

## Figures and Tables

**Figure 1 antioxidants-12-01248-f001:**
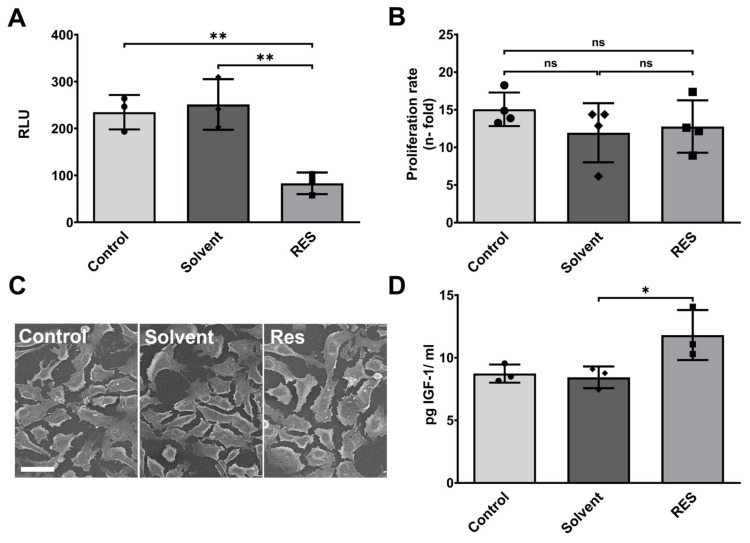
Resveratrol rapidly reduces inflammasome activity and induces IGF-1 secretion without influencing cell proliferation or morphology. (**A**) Caspase-1-activity-based inflammasome luminescence assay shows that the 6 h resveratrol treatment decreases inflammasome activity. (**B**) The 100 µM resveratrol treatment does not have any significant effect on cell growth over 24 h. Proliferation rate calculated as an n-fold increase in the initially seeded cell number. (**C**) Representative scanning electron images (SEM) (scale bar: 50 µm) display a similar cell morphology after 24 h. (**D**) ELISA of IGF-1 concentrations in cell supernatant reveals higher IGF-1 concentrations after the 24 h resveratrol treatment. HMC3 was co-treated with/without 100 µM resveratrol (RES) and compared to the untreated control. Solvent = PEG400; RLU = relative light unit; n = 3/4 number of independent cell cultures; ns = not significant, * *p* < 0.05 and ** *p* < 0.01.

**Figure 2 antioxidants-12-01248-f002:**
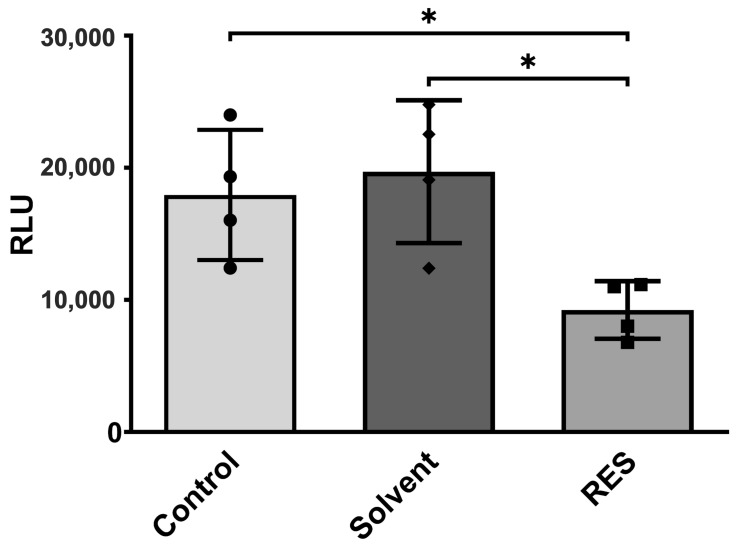
Lower glucose consumption after resveratrol treatment. Glucose uptake luminescence assay indicated a reduced glucose consumption in samples treated with resveratrol. HMC3 was co-treated with/without 100 µM resveratrol (RES) for 24 h and compared to untreated control. Solvent = PEG400; RLU = relative light unit; n = 4 number of independent cell cultures. * *p* < 0.05.

**Figure 3 antioxidants-12-01248-f003:**
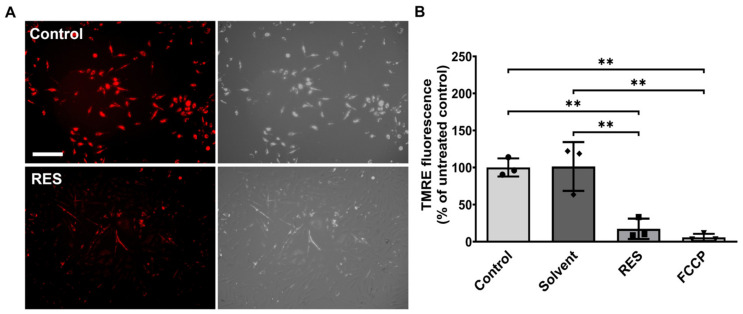
Resveratrol decreases mitochondrial activity. (**A**) Representative fluorescence and brightfield images of TMRE staining (scale bar: 100 µm) showed less TMRE accumulation in resveratrol-treated samples. TMRE accumulates in active, negatively charged mitochondria. (**B**) Corresponding quantification of TMRE fluorescence intensity in percentage normalized relative to the untreated control. HMC3 was co-treated with/without 100 µM resveratrol (RES) for 24 h and compared to the untreated control. Solvent = PEG400; FCCP = carbonyl cyanide 4-(trifluoromethoxy) phenylhydrazone; n = 3 number of independent cell cultures. ** *p* < 0.01.

**Figure 4 antioxidants-12-01248-f004:**
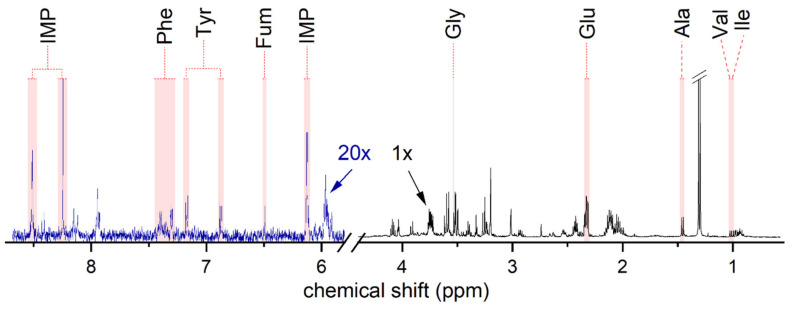
Representative 500 MHz ^1^H NMR spectrum of untreated human HMC3 microglia cells in the control group (2342 acquisitions). Metabolites were identified regularly and quantified with respect to the total metabolite signal. The intensity of the spectrum above 5.8 ppm, also indicated by the blue color, is enhanced by a factor of 20 for better visibility of resonance lines. The region of 4.2–5.8 ppm was affected by water suppression and is not shown. Ala = Alanine; Fum = fumarate; Glu = glutamate; Gly = glycine; IMP = inosine monophosphate; Ile = isoleucine; Phe = phenylalanine; Tyr = tyrosine; Val = valine.

**Figure 5 antioxidants-12-01248-f005:**
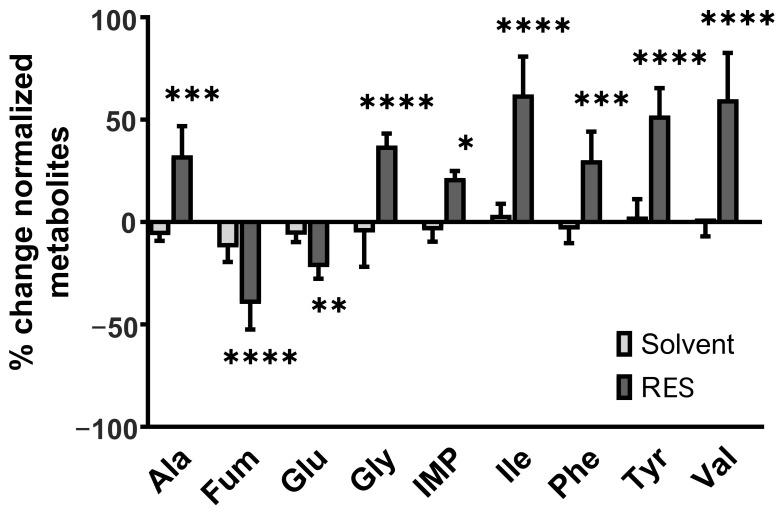
Relative changes in selected metabolites of human HMC3 microglia cells following 24 h of resveratrol treatment (100 mM). Normalized metabolite changes (percentage of corresponding controls) in human HMC3 microglia cells. HMC3 was co-treated with/without 100 µM resveratrol (RES) for 24 h and compared to the untreated control. Solvent = PEG400; n = 3 number of independent cell cultures. * *p* < 0.05; ** *p* < 0.01; *** *p* < 0.001; **** *p* < 0.0001.

## Data Availability

The data that support the findings of this study are available from the corresponding author upon reasonable request.

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
