# Peer review of "Resveratrol Mitigates Metabolism in Human Microglia Cells"

_antioxidants, 2023, doi:10.3390/antiox12061248_

Round 1
Reviewer 1 Report
The manuscript under review entitled “Resveratrol mitigates metabolism in human microglia cells” is interesting since it aims to shed light on the protective action of resveratrol in the neuroinflammatory state. The authors propose to verify the ability of this polyphenol to modulate the endogenous inflammasome activation and this may have some interest for those studying the action of resveratrol in neuroinflammation and neurodegeneration.
After all, I have some observation to do on this work and some changes are need before a possible publication of this manuscript:
- Why did the Authors choose the concentration of 100 micromoles of resveratrol to treat human microglia? Are there already references in the literature of this concentration used on human microglia? It would be appropriate for the Authors to include a reference.
- The Authors highlight in this work that resveratrol is able to determine a switch to an anti-inflammatory M2 phenotype of human microglia cells by mitigating metabolism, decreasing endogenous inflammasome activity and increasing IGF-1 production. The analyses performed by the Authors and the observed cellular responses are certainly interesting, however, a phenotypic typing of the M1/M2 shift of microglia due to resveratrol is lacking. The authors should add an experiment in which they show the modulation of markers of the M1 phenotype and M2 phenotype.
- Line 291: A reference should be added for example on the ability of resveratrol to inhibit the production of pro-inflammatory cytokines or increase that of anti-inflammatory proteins in microglia exposed to LPS.
Overall, I recommend publication of this manuscript after these minor changes have been made.
Author Response
We thank the reviewer for very valuable comments. Please see the attachemnt for our point-to-point responses.

Reviewer 2 Report
The manuscript (antioxidants-2442182) evaluated the effects of resveratrol on human HMC3 microglia cells. Resveratrol at 100 μM for 6 hr or 24 hr was found to suppress the activity of caspase-1, an index of inflammasome activity, increase the release of IGF-1, decrease glucose uptake, inhibit mitochondrial activity, and attenuate cellular metabolism in a 1H NMR assay of whole cell lysate. It was concluded that resveratrol supports the switch to an anti-inflammatory M2 phenotype of human HMC3 microglia cells by mitigating metabolism, decreasing endogenous inflammasome activity and increasing IGF-1 production.
Overall, the study was well executed and the study of mitochondrial activity and metabolism in microglia by resveratrol appears to be novel. Perhaps the biggest issue with the reviewer is the emphasis of M1/M2 switch by resveratrol in the manuscript as the phenotype of microglia was not really characterized in the study. Therefore, it is an overstretch to say that resveratrol “promotes the anti-inflammatory M2 microglia phenotype” in the manuscript. I have the following more detailed comments for the authors.
1. Abstract should be re-written. It should summarize the major rationale, methods, results and conclusion supported by the findings instead of just generic description.
2. Resveratrol was used at a high concentration of 100 μM. Although this is a commonly used condition, it is far from in vivo levels of the drug that can be possibly achieved, e.g., in low μM or sub-μM concentrations. Is this why many fail in clinical trials? Can the authors comment on it, in light of dose-dependent effects of resveratrol, e.g., Mukherjee et al. Dose Response. 2010; 8:478-500; Madreiter-Sokolowski et al. Nutrients 2017, 9, 1117?
3. Are immortalized cultured microglia already in an abnormally ‘activated’ state in vitro that is far from M1/M2 polarization? Given the controversy around microglial polarization, is it useful to even use the M1/M2 terms in this manuscript?
4. The effect on mitochondria shown in Figure 3 is astonishing. Did the authors examine mitochondrial and cell morphology? How about a dose response? The other findings would not be surprising given the scale of mitochondrial shutdown. In this respect, resveratrol looks like a mitochondrial toxin with cytotoxicity as in many cancer cell lines. Are the microglia under the resveratrol treatment conditions in a quiescent state or …?
Author Response
We thank the reviewer for very valuable comments. Please see the attachment for our point-to-point responses.
